The experience of a tele-operated avatar being touched increases operator’s sense of discomfort

Kimoto Mitsuhiko kimoto@atr.jp
Shiomi Masahiro
Interaction Science Laboratories, ATR , Seika-cho , Kyoto , Japan
Fontana Simone
Electronic publication date: 2024 Mar 19
Publication date: 2024
Volume: 10
Electronic Location ID: e1926
Received 2023 Nov 22; Accepted 2024 Feb 13
Copyright: ©2024 Kimoto and Shiomi
Copyright year: 2024
Copyright holder: Kimoto and Shiomi
License: This is an open access article distributed under the terms of the Creative Commons Attribution License, which permits unrestricted use, distribution, reproduction and adaptation in any medium and for any purpose provided that it is properly attributed. For attribution, the original author(s), title, publication source (PeerJ Computer Science) and either DOI or URL of the article must be cited.
License URL: https://creativecommons.org/licenses/by/4.0/

Keywords: Human–computer interaction, Avatar, Tele-operated system, Virtual agent, Social touch

Funding: JST Moonshot R&D JPMJMS2011 JSPS KAKENHI JP23K16980 This research work was supported by JST Moonshot R&D Grant Number JPMJMS2011 (system development, experiment, writing) and JSPS KAKENHI Grant Numbers JP23K16980 (analysis). There was no additional external funding received for this study. The funders had no role in study design, data collection and analysis, decision to publish, or preparation of the manuscript.

==============================
Recent advancements in tele-operated avatars, both on-screen and robotic, have expanded opportunities for human interaction that exceed spatial and physical limitations. While numerous studies have enhanced operator control and improved the impression left on remote users, one area remains underexplored: the experience of operators during touch interactions between an avatar and a remote interlocutor. Touch interactions have become commonplace with avatars, especially those displayed on or integrated with touchscreen interfaces. Although the need for avatars to exhibit human-like touch responses has been recognized as beneficial for maintaining positive impressions on remote users, the sensations and experiences of the operators behind these avatars during such interactions remain largely uninvestigated. This study examines the sensations felt by an operator when their tele-operated avatar is touched remotely. Our findings reveal that operators can perceive a sensation of discomfort when their on-screen avatar is touched. This feeling is intensified when the touch is visualized and the avatar reacts to it. Although these autonomous responses may enhance the human-like perceptions of remote users, they might also lead to operator discomfort. This situation underscores the importance of designing avatars that address the experiences of both remote users and operators. We address this issue by proposing a tele-operated avatar system that minimizes unwarranted touch interactions from unfamiliar interlocutors based on social intimacy.

Introduction

People are experiencing increased opportunities to interact remotely by tele-operated on-screen computer graphics (CG) characters and robots, which we label tele-operated avatars. These systems are dissolving spatial and physical constraints, enabling such interactions and assistance as guided tours in museums (Roussou et al., 2001) and customer service in stores (Takeuchi, Yamazaki & Yoshifuji, 2020; Song et al., 2022; Barbareschi et al., 2023b).

Research into tele-operated avatars has typically been conducted from two perspectives: the operators and the remote users with whom they interact. From the former perspective, a variety of methods and systems have been proposed to enhance the control of the system and accurately convey the events occurring in remote locations. For instance, Barbareschi et al. (2023b) developed systems that enable people to simultaneously control multiple robotic avatars to support more accessible multitasking styles. Tanaka, Takahashi & Morita (2013) proposed a tricycle-style operation interface that enables children to control a telepresence robot. From the perspective of remote users, various studies have identified the effects of autonomous and semi-autonomous behaviors of tele-operated avatars on the impressions of remote users. For example, autonomous nodding behaviors by robots increase the social presences of a remote operator (Tanaka et al., 2016). Tanaka et al. (2019) proposed an android robot system that autonomously focused on speakers and looked at them. This body of research has improved the controllability of tele-operated avatars and enhanced the positive impressions left on remote users. Essentially, such studies are centered around remote locations, focusing on the interactions between tele-operated avatars and the people with whom they are engaging.

When designing avatar systems, note that the nature of the interaction by tele-operated avatars often differs from traditional face-to-face communication. Touching is particularly more common when interacting with tele-operated avatars. These avatars are frequently displayed on touchscreens or have a touchscreen like Pepper (developed by SoftBank Robotics) that allows graphical user interface (GUI) selection menus to be provided for their services. Although such touch-inducing interfaces may also increase the touching of avatars, touch interaction with strangers is not typical in face-to-face interactions (Suvilehto et al., 2015; Suvilehto et al., 2019). Past research reported that a robot’s reaction to being touched is crucial to leave natural, human-like impressions on people, and the autonomous reaction behavior design of CG agents and robots has been proposed (Shiomi et al., 2018; Mejía et al., 2021; Kimoto et al., 2023). These studies indicate the importance of agents’ reactive behaviors to human touch in conveying human-like impressions to remote users. In avatar systems, the haptic sensations are often utilized to control avatars intuitively and to understand remote environments (Salisbury, Conti & Barbagli, 2004; Colella et al., 2019; Ho & Nakayama, 2021; Tanaka et al., 2022; Lenz & Behnke, 2023). These studies mainly focus on reproducing realistic haptic sensations for the avatar’s operator.

Past research has insufficiently assessed the perspective of operators who receiving haptic feedback from touch interactions at remote locations. Consequently, the impact of touch interactions on operators between a remote user and an avatar has inadequately been investigated. This study investigates whether operators experience the perception of being touched when a tele-operated avatar is physically interacted with in a remote location, particularly when the avatar displayed on the screen is touched. Specifically, we aim to clarify two types of perceptions: the perception of touch experienced by the operator and the operator’s perception of the avatar. To measure these perceptions, we use a series of questionnaires. These questions cover the operator’s feeling of touch when the avatar is touched, any discomfort felt, their ability to concentrate during the interaction, the ease with which they can sense when the avatar is touched, the sense of ownership they feel over the avatar’s body, their perception of control over the avatar’s movements, and their likeability of the avatar.

The structure of this article is as follows: ‘Related Work’ provides a detailed description of related research and outlines the position of this study. ‘Materials & Methods’ explains the tele-operated avatar systems used to explore the research questions and details the experimental setup. ‘Results’ presents the results of the experiments. In ‘Discussion’, we discuss the implications arising from the results related to the operator’s perceptions of interaction and offer suggestions for avatar system design. Finally, ‘Conclusions’ summarizes the article as a whole.

Related Work

Tele-operated avatars

The role of tele-operated avatars, encompassing both tele-presence robots and CG characters on monitors, is gaining traction for enhancing remote communication. These avatars enable individuals to interact and offer services without requiring a physical presence. Both a robot in a physical space and an avatar on a screen support social interactions between people across distances and help them overcome challenges related to scheduling, disabilities, or other accessibility issues (Zhang & Hansen, 2022). Takeuchi, Yamazaki & Yoshifuji (2020) proposed the concept of “avatar work”, which is a style of telework that enables people with disabilities to engage in service provisions in physical environments. They developed robots for their new telework concept and opened an avatar robot café where disabled people operate them.

Numerous studies have explored different design facets for user-friendly and approachable hardware and interfaces (Colella et al., 2019; Schwarz et al., 2021; Dafarra et al., 2022; Erp et al., 2022; Zand, Ren & Arif, 2022; Barbareschi et al., 2023a; Behnke, Adams & Locke, 2023; Lenz et al., 2023). At the ANA Avatar XPRIZE competition, many robotic avatar systems were developed and evaluated based on their support for remotely interaction with humans (Behnke, Adams & Locke, 2023). NimbRo, the prize-winning avatar system (Schwarz et al., 2021; Schwarz et al., 2023; Lenz et al., 2023) is capable of remotely interacting with humans and performing such complex tasks as jointly solving a puzzle, exploring an artifact by touching it, and using a drill. Barbareschi et al. (2023a) presented a parallel control system allowing disabled workers in a café to embody multiple robotic avatars at the same time to carry out different tasks. Zhang, Hansen & Minakata (2019) proposed a mobile telepresence robot that can be controlled by the eye-gaze movements of people with motor disabilities. Other studies of avatar systems have focused on the perceptions of interacting partners at remote locations (Tanaka et al., 2016; Tanaka et al., 2019; Baba et al., 2021; Ijuin et al., 2021; Chung & Jo, 2022; Pakanen et al., 2022; Song et al., 2022). Chung & Jo (2022) proposed a method that aligns the gaze directions of a virtual avatar with those of a person at a remote location. Their proposed method improved the co-presence perceptions of people at remote locations with an avatar. Baba et al. (2021) studied how different forms of social presence in a tele-operated robot affect customer interactions in a supermarket. They concluded a “costume” form (with an operator’s photo and voice conversion) was the most effective in terms of performance, as measured by the stopping rate of people, the conversation rate with the robot, and the number of accepted flyers.

These studies have provided knowledge for creating immersive experiences that allow operators to control tele-operated avatars, improving user perceptions in remote locations. Unfortunately, focus is limited on how interactions between people and avatars in remote locations influence the operator behind the latter.

Haptic feedback in avatar systems

The haptic sensation is fundamental feedback to humans for understanding and interacting with their surroundings (Robles-De-La-Torre, 2006). To achieve immersive and intuitive control of avatars, various devices and mechanisms have been proposed to communicate haptic sensations to people in different locations (Salisbury, Conti & Barbagli, 2004; Colella et al., 2019; Ho & Nakayama, 2021; Tanaka et al., 2022; Lenz & Behnke, 2023). For example, Colella et al. (2019) proposed a device that offers proprioceptive information of the opening of an artificial hand through sensory substitution, specifically by longitudinal skin stretch stimulation. Lenz & Behnke (2023) introduced a bimanual telemanipulation system that blends an anthropomorphic avatar robot with an operator station that delivers force and haptic feedback for enhancing immersion and task efficiency in robotic teleoperation.

Moreover, while haptic devices provide direct physical feedback to users, interest exists in understanding how visual feedback can be leveraged to convey haptic interactions. Many studies have confirmed that it can evoke and leverage haptic sensations. This phenomenon is referred to as “pseudo-haptic” (Lécuyer, 2009; Pusch & Lécuyer, 2011; Ujitoko & Ban, 2021). In other words, even without actual tactile stimuli, certain visual cues can evoke sensations similar to haptic feedback. For instance, Costes et al. (2019) proposed symbolic cursor effects for touchscreens and showed that the effects evoke perceptual dimensions like hardness, friction, and roughness through changes in the cursor’s shape and motion. Kim & Xiong (2022) introduced and assessed four pseudo-haptic features (proximity feedback, protrusion, hit effect, and penetration blocking) that aim to simulate haptic sensations without any actual physical stimuli. Through a user study that interacted with buttons into which these features were integrated, they determined that all the features significantly enhanced the user experience across various dimensions, including haptic illusion, satisfaction, and sense of reality. More closely related to avatar systems, Aymerich-Franch et al. (2017) investigated the haptic sensations of operators using robot avatars. In their experiments, operators reported haptic sensations in their real hands when their robot avatar touched a curtain.

Numerous studies have developed techniques and explored the pseudo-haptic phenomenon to provide a more realistic haptic interaction experience that closely resembles physical environments. These findings underscore the significance of haptic sensations in avatar systems when operators touch objects to interact with remote surroundings. However, it remains unclear how such haptic feedback, meant to enhance haptic sensations, impacts operators when avatars are touched by users at remote locations. In other words, the effects on the operators of interactions involving touch between remote users and avatars are still unknown.

Materials & Methods

Task design

We designed a task that supposed a situation where an operator talked with an interlocutor at a remote location by tele-operating a CG agent. From the remote location side, the interlocutor talked with the tele-operated CG agent displayed on a monitor. When the operator was providing information, the interlocutor touched the agent displayed on a monitor. After the explanations, we evaluated the operator’s perceptions by questionnaires.

System

We have developed a tele-operated avatar system for an operator to talk with an interlocutor on a remote location through a CG character (called the operator’s avatar). To explore the effects on the operators of touch interaction between remote users and avatars, we employed a system design that provides visual feedback without physical haptic feedback. This approach was chosen because most avatar systems rely primarily on visual feedback.

Figure 1 shows an overview of interactions implemented through our developed tele-operated avatar system and the data flow of video streaming. The operator can move their avatar, which is located in a virtual environment. The area in front of the display in the virtual environment is streamed on the monitor at the remote location. The virtual environment’s display plays real-time stream video of the remote locations. When the operator moves their avatar to the area in front of the display, their avatar is streamed on the monitor at the remote location. The operator moves the avatar to the area in front of the monitor where it provides information and talks with the interlocutor. The avatar is lip-synced with the operator whose interface is divided into three panels: monitor, interlocutor, and avatar. The monitor panel, which is used to watch the video stream on the remote space, also controls the avatar’s locations. The system’s view selector module, controlled by the experimenter, determines the view types presented to the operator: a first-person view of the avatar (Fig. 2B), a third-person view of the avatar (Fig. 2C), or a direct view from the video stream of the remote location (Fig. 2A). These views are designed to evaluate the effects of the operator’s perspective in experiments. The interlocutor panel is used to watch the video stream of the face of the interlocutor who is talking with the operator. The avatar panel shows the current status of the avatar and confirms its status, e.g., moving or talking. On the remote location side, the interlocutor can talk in front of the monitor that streams the operator’s avatar in the virtual environment.

Figure 1 Overview of interaction implemented by developed tele-operated avatar system and data flow of video streaming.

Figure 2 Examples of three types of visual perspectives displayed on monitor panel.

(A) Video streaming of remote location. (B) First-person view of avatar. (C) Third-person view of avatar.

Figure 3 presents the system architecture of the tele-operated avatar system developed for this study. It is comprised of four primary modules: touch recognition, touch visualization, view selector, and avatar control. The touch recognition module identifies the specific facial or bodily points at which the avatar is touched. This identification is based on sensory input from the touch monitor at a remote site. The touch visualization module visually emphasizes the physical contact by the remote user (Fig. 4). It uses a “hit effect” to visually emphasize the point of contact. This effect has successful induced pseudo-haptic sensations (Kim & Xiong, 2022). Furthermore, the module prompts the avatar’s autonomous reaction behavior through which it bends backward in response to being touched, conveys a sense of touch to the operator, and offers a more natural impression to the remote user. The view selector module allows experimenters to choose the visual perspective displayed on the monitor panel, determining what the operator sees. The avatar control module manages the avatar’s movement. The avatar lip-syncs based on the operator’s speech features. Additionally, operators can move the avatar using a joystick on a game controller. The video stream between the remote location and the operator’s side was streamed using the Real-Time Streaming Protocol (RTSP), with a delay of approximately 500 ms. Regarding the specifics of the devices used, we employed a BenQ GL2450-T monitor on the operator’s side and an I-O DATA LCD-MF224FDB-T as the touch monitor on the remote side. The remote interlocutor was captured using a Logicool C920t network camera and streamed to the interlocutor panel, while the remote environment was captured using an I-O DATA TC-PC8Z network camera and streamed to the monitor panel. For audio communication, we used Zoom software, and Sennheiser SC 165 headsets were used by both the operator and the interlocutor.

Figure 3 System architecture of developed tele-operated avatar system.

Figure 4 Avatars used and examples of their reaction behaviors.

Hypothesis and prediction

In this study, we focus on the perception of touch experienced by the operator and the operator’s perception of the avatar during touch interactions. Although the operator is not equipped with any device for haptic feedback, related works have suggested that visual feedback can evoke perceptions of pseudo-touch (Lécuyer, 2009; Pusch & Lécuyer, 2011; Aymerich-Franch et al., 2017; Ujitoko & Ban, 2021). Additionally, virtual reality research has shown that social touch interactions can induce emotional reactions in users who are controlling avatars (Huisman et al., 2014; Erp & Toet, 2015; Sykownik & Masuch, 2020). However, despite these foundational studies, a direct connection between a remote user touching a displayed avatar and inducing pseudo-touch perceptions in its operator has not been explicitly explored. Based on these considerations, we make the following prediction: A remote user’s touch on a displayed avatar will evoke perceptions of pseudo-touch in its operator.

Conditions

We considered the following factors: (1) touch visualization and (2) the operator’s user interface (UI). Both touch visualization and perspective were treated as within-subject factors, resulting in each participant experiencing six conditions that combined two levels of touch visualization with three levels of perspective.

(1) Touch visualization (two levels: with/without, within-subject): We investigated the effects by using and not using the touch visualization module. In the “with” level, both the hit effect and the avatar’s reaction were present. In the “without” level, both visual cues were absent.

(2) Operator’s UI (three levels: first-person/third-person/direct-view, within-subject): We investigated the type of perspective from which the experimental participants operated the avatar (Fig. 2). We prepared a first-person perspective (Fig. 2B), a third-person perspective (Fig. 2C), and a direct view of the remote location (a direct perspective) (Fig. 2A).

Procedure

The participants first received explanations about the experiment overview and filled in a consent form. In these explanations, participants were informed about the possibility that their avatar could be touched by an experimenter at a remote location. Next they used the experimental system to control an avatar and provided answers to the experimenter in another room. Participants were given the following sentences as a template for the information explanation and provided responses based on a selected topic: “My favorite (topic) is ‘***.’ I like it because it’s ‘***.’ I’ve liked it since ‘***’.” Six items were prepared to fit the topic: food, hobbies, movies, tourist spots, sports, and animals. At the start of explaining to the participants each of the three contents, the experimenter simultaneously touched one of three distinct locations on the participant’s avatar that was displayed on the screen: the face, the right shoulder, or the left shoulder. Each location was touched only once, and the order of the touches was randomized throughout the explanations. After explaining each topic, the participants filled out a questionnaire. They repeated this procedure six times, combining two levels of touch visualization factors and three levels of perspective factors. The combinations of the factors and topics and the order in which they were presented were determined randomly. We prepared two types of avatars: a man and a woman (Fig. 4). Half of the participants used avatars matching their own gender, while the other half used avatars of a different gender. All the procedures were approved by the ATR Review Board Ethics Committee (501-4).

Participants

Forty participants joined the study: 20 men and 20 women, all of whom were native Japanese speakers (mean age = 29.9 years, SD = 6.77).

Measurement

To evaluate the touch perceptions of the operator, we prepared the following questionnaire items, referring to related studies (Kalckert & Ehrsson, 2012; Matsumiya, 2021). The items were evaluated in a 7-point response format (where 1 is the most negative and 7 is the most positive). The following are the questionnaire items; their labels are shown in brackets:

Q1. I felt as if my own body was being touched when the avatar’s body was touched. (touch perception)

Q2. I felt discomfort when the avatar’s body was touched. (discomfort)

Q3. I was able to maintain my concentration while I was explaining. (focus)

Q4. It was easy to sense when my avatar was being touched. (understandability)

Q5. I felt as if the avatar’s body was my own body. (ownership)

Q6. I felt as if the avatar moved as if obeying my will. (agency)

In addition, to evaluate the avatar’s perceptions, we prepared questionnaires items to assess intention to use (Heerink et al., 2008) and the avatar’s likeability (Bartneck et al., 2009). Intention to use is a scale designed to assess technology acceptance. It was composed of three questionnaire items in a 7-point response format: (1) I think I’ll use *** in the next few days, (2) I am certain to use *** in the next few days, and (3) I’m planning to use *** in the next few days (in our experiment, *** was filled with the tele-operated system). Likability is a scale used to assess positive impressions of the target and was composed of five questionnaire items with 5-point semantic differential scales: dislike/like, unfriendly/friendly, unkind/kind, unpleasant/pleasant, and awful/nice. The final scores for intention to use and likability were calculated as the mean scores of the respective questionnaire items.

Environment

Figure 5 illustrates the experimental environments. The participants controlled the displayed CG agents and provided answers to the experimenter who listened to the conversations of the participants and touched the displayed agent from another room.

Figure 5 Experimental environments.

Results

Figure 6 shows the questionnaire results. We conducted a two-way repeated measure ANOVA for the touch visualization and the operator’s UI factors.

Figure 6 Results of questionnaire items: error bars represent standard error.

*: p < 0.05, **: p < 0.01.

For Q1, touch perception, we found no significant effects of the touch visualization factor (F (1, 39) = 2.376, p = 0.131, ηp2 = 0.057), the operator’s UI factor (F (2, 78) = 0.233, p = 0.793, ηp2 = 0.006), or the interaction between the touch visualization and the operator’s UI factors (F (2, 78) = 0.170, p = 0.844, ηp2 = 0.004).

For Q2, discomfort, we found significant effects of the touch visualization factor (F (1, 39) = 11.194, p = 0.002, ηp2 = 0.223). We found no significant effects of the operator’s UI factor (F (2, 78) = 0.419, p = 0.659, ηp2 = 0.011) or the interaction between the touch visualization and the operator’s UI factors (F (2, 78) = 0.160, p = 0.852, ηp2 = 0.004).

For Q3, focus, we found no significant effects of the touch visualization factor (F (1, 39) = 4.0684, p = 0.051, ηp2 = 0.094), the operator’s UI factor (F (2, 78) = 0.0622, p = 0.940, ηp2 = 0.002), or the interaction between the touch visualization and the operator’s UI factors (F (2, 78) = 0.0852, p = 0.918, ηp2 = 0.002).

For Q4, understandability, we found significant effects of the touch visualization factor (F (1, 39) = 10.524, p = 0.002, ηp2 = 0.213). We found no significant effects of the operator’s UI factor (F (2, 78) = 0.115, p = 0.892, ηp2 = 0.003) or the interaction between the touch visualization and the operator’s UI factors (F (2, 78) = 1.411, p = 0.250, ηp2 = 0.035).

For Q5, ownership, we found no significant effects of the touch visualization factor (F (1, 39) = 0.0534, p = 0.818, ηp2 = 0.001), the operator’s UI factor (F (2, 78) = 0.3636, p = 0.696, ηp2 = 0.009), or the interaction between the touch visualization and the operator’s UI factors (F (2, 78) = 2.6407, p = 0.078, ηp2 = 0.063).

For Q6, agency, we found significant effects of the operator’s UI factor (F (2,78) = 4.6337, p = 0.013, ηp2 = 0.106). Multiple comparisons with Tukey’s HSD for the operator’s UI factor revealed significant differences between the conditions: third-person >direct-view (p = 0.014). We found no significant effects of the touch visualization factor (F (1, 39) = 0.0576, p = 0.812, ηp2 = 0.001) or the interaction between the touch visualization and the operator’s UI factors (F (2, 78) = 0.7163, p = 0.492, ηp2 = 0.018).

For intention to use, we found no significant effects of the touch visualization factor (F (1, 39) = 1.784, p = 0.189, ηp2 = 0.044), the operator’s UI factor (F (2, 78) = 0.420, p = 0.659, ηp2 = 0.011), or the interaction between the touch visualization and the operator’s UI factors (F (2, 78) = 2.144, p = 0.124, ηp2 = 0.052).

For likeability, we found no significant effects of the touch visualization factor (F (1, 39) = 0.9341, p = 0.340, ηp2 = 0.023), the operator’s UI factor (F (2, 78) = 2.7803, p = 0.068, ηp2 = 0.067), or the interaction between the touch visualization and the operator’s UI factors (F (2, 78) = 0.0719, p = 0.931, ηp2 = 0.002).

These results suggest that emphasizing touch interactions in remote locations enhances the understandability of being touched and heightens the operator’s discomfort when being touched. The third-person view also seems to increase the agency of the avatar’s movement more than the direct view. The avatar’s autonomous behaviors and system interface, which highlight touch, did not directly induce pseudo-touch sensations in the operator. However, they did evoke the emotional perceptions of pseudo-touch, manifesting as discomfort, thereby partially supporting the prediction.

Discussion

Implications

The results of our experiments indicate that emphasizing touch from a remote user increases the operator’s discomfort, a result that suggests the subtlety of managing touch interactions with remote users. These interactions must balance the demands of social engagement with the preservation of personal boundaries. Such boundaries will ultimately vary depending on the nature of the touch and the relationship between remote users and operators. Thus, these findings imply that tele-operated agents should minimize unnecessary touch to maintain operator comfort.

Presenting remote situations as clearly as possible helps the operator understand her remote environment and any interactions taking place. While autonomous reactions of avatars can enhance the operator’s perceptions, our findings also showed that such touch interactions can cause negative feelings in the operator. When designing tele-operated avatar systems, the impact of interactions must be considered between remote users and avatars on the operators to ensure a balanced experience for both parties. In our task, the perceptions of pseudo-haptics elicited negative feelings in the operators. However, in tasks where touch is perceived positively, such as a social touch with a partner or playing with a pet, the phenomena observed in our study can enhance positive experiences for operators.

Tele-operated agent refusing excessive touch

Our experimental results suggest that tele-operated agents should not always accept being touched by remote users. As such an avatar system design, it is conceivable to take into account that the topography of social touch enables the avatar to autonomously avoid unnecessary touching. As delineated by Suvilehto et al. (2015); Suvilehto et al. (2019), this concept is based on findings that identify the specific bodily regions where each person in their social network is permitted to touch. Figure 7 presents an idea of system architecture for implementing the idea of the social touch based on the system (Fig. 3) used for our experiments. A unique feature of the concept system is the social touch response module, which manages the reaction to being touched by a remote user, is based on the touched bodily region and the social relationships between the remote users and operators. When the agent is touched by a remote user, the system calculates the touch tolerance using a preset touch tolerance matrix, which accounts for various combinations of bodily parts (hands, shoulders, and face) and social relationships (including stranger, friend, or partner). Our system calculates touch tolerance based on this preset matrix each time the agent is touched by the remote user. The matrix draws upon established touch norms from social touch studies (Suvilehto et al., 2015; Suvilehto et al., 2019; Kimoto et al., 2023). For instance, these studies suggest that strangers are permitted to touch only the hand, while friends can also touch the shoulder, which is the second most permissible area. After being touched, the system adds a discomfort value, which corresponds to the touched bodily part and the remote user’s relationship (denoted as ΔD = MRB) to the accumulated discomfort value. Term MRB represents the discomfort value in the matrix for the specific combination of relationship R and bodily part B. Accumulated discomfort value D is then updated using an equation :D = D + ΔD. Our system also maintains a maximum touch tolerance level, represented as T, which is a predetermined constant. If accumulated discomfort value Dexceeds touch tolerance level T, the avatar automatically refuses the touch and states, “Please don’t touch me”. Formally, this condition is stated as: if D > T, the system refuses a touch, ensuring that the tele-operated avatar interacts in a way that respects both the social norms and the operator’s comfort boundaries inherent to touch.

Figure 7 System architecture of tele-operated avatar system that refuse unnecessary touch.

Limitations

Our experiment was designed based on a scenario where an operator provides information to users in remote locations. Thus, depending on the task and the interaction, the observed perceptions may vary. Furthermore, participants were informed that their avatar could be touched, making the touch interactions predictable for them. Had these interactions been unexpected, they might have induced stronger perceptions than those observed in our experiment. Although the visual feedback in our system is limited, we utilized the avatar’s reaction behaviors and hit effects, as validated in related works. However, the optimal feedback to elicit pseudo-haptics depends on specific interactions. Moreover, we did not examine how to leverage touch sensations and pseudo-haptics in interactions within the remote user-avatar-operator triad. Therefore, our future research will address the effects observed in this study.

In the experiments, 40 people aged between 20 and 50 years (mean age = 29.9, SD = 6.77) participated. We did not control for the distribution of age; therefore, we cannot conclude that the indications from the results are applicable to specific generations. For example, technology acceptance and habituation to tele-operated systems may vary across generations (Hauk, Hüffmeier & Krumm, 2018), and this could affect the results. In addition, we did not investigate the effects of factors related to social interactions. For example, several studies have reported the impact of gender in the fields of social touch and proxemics (Rivu et al., 2021; Zibrek et al., 2022; Kimoto et al., 2023), suggesting that gender effects should be further explored in the context of an operator’s perceptions. The cultural influences on the acceptability of social touch have also been investigated in various works (Suvilehto et al., 2019; Burleson et al., 2019). In our study, all participants reported that they were native Japanese, and thus the cultural effects were not clear in our experiment. Touch styles may also affect the perceptions of operators. Some studies have reported that the touch styles of robots are related to their social acceptance (Zamani et al., 2020; Okada et al., 2022) or emotional expressions (Zheng et al., 2021). Future studies focusing on various social effects would be beneficial in providing deeper insights and more nuanced suggestions.

Conclusions

We reported the results of an experimental investigation into the perceptions of being touched that arise in operators when a tele-operated avatar displayed on the screen is touched by an interlocutor at a remote location. Our focus was on two types of perceptions: the perception of touch as experienced by the operator and the operator’s overall perception of the avatar. Our findings indicate that even if the touch is directed at an avatar displayed on a screen, the touch’s visualization through the hit effect and automated responsive movements can evoke the emotional perceptions of being touched as creating discomfort. Moreover, this perception of non-contact occurs regardless of the perspective from which the avatar is operated. Although automation of the avatar’s movements is expected to contribute to the naturalness and human-like impressions of avatar from the perspective of remote interlocutors, the visualization of being touched may cause discomfort in operators. This possibility highlights the need to evaluate and design avatar movements from the perspectives of both remote interlocutors and avatar operators.

Supplemental Information

Supplemental Information 1 Anonymized data set and questionnaire items

Supplemental Information 2 Questionnaire

Additional Information and Declarations

Competing Interests

Author Contributions

Ethics

Data Availability

The authors declare there are no competing interests.

Mitsuhiko Kimoto conceived and designed the experiments, performed the experiments, analyzed the data, performed the computation work, prepared figures and/or tables, authored or reviewed drafts of the article, and approved the final draft.

Masahiro Shiomi conceived and designed the experiments, authored or reviewed drafts of the article, and approved the final draft.

The following information was supplied relating to ethical approvals (i.e., approving body and any reference numbers):

All the procedures were approved by the ATR Review Board Ethics Committee (501-4).

The following information was supplied regarding data availability:

The anonymized raw data set and questionnaire items are available in the Supplemental Files.

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
