# Peer review of "The experience of a tele-operated avatar being touched increases operator’s sense of discomfort"

_PeerJ Computer Science, doi:10.7717/peerj-cs.1926_

## Round 0.1 · original submission · Minor Revisions

The reviewers agree on the quality of the paper and on the contribution of the authors to this particular research topic. However, there are minor issues, mainly regarding the presentation and the description of the experimental setup, that should be fixed before the paper is ready for publication. Please refer to the reviews for the details.

Reviewer 1 ·

Basic reporting

The text is well written, clear, and unambiguous. There are a few minor typos, particularly in the use of the pronoun "her" instead of "their". An example is in line 163, where the operator's avatar is referred to with the pronoun "her", whereas in line 166 it is referred to with the pronoun "their". I suggest the authors use the pronoun "their" instead of "her" in order to achieve gender neutrality. In addition, in line 209, the article “A” after the colon should be lowercase. Finally, some acronyms are used before they are defined in the text. Examples are: CG in line 36, and GUI in line 59.

The previous literature about the topic of the analysis is well detailed and described in the “Introduction” and “Related Work” sections. One part of the description that could be improved is that in lines 66-67, where the lack of literature on the operator's perspective is underlined. This concept, that is at the base of your work, is well described in the section “Haptic Feedback in Avatar Systems” but it is poorly described in the Introduction where no literature seems to exist on the topic. I suggest that the authors integrate this part as they have already done for the works related to the avatar controls by the operators or the importance of robot (or avatar) reaction in conveying human-like impressions to remote users.

The manuscript's figures are clear and visible, but some of them could be more detailed. For instance, in Figure 1, the authors should better specify the position of the monitor panel, the interlocutor's panel, and the avatar panel. Additionally, the position of the Experimenter subject in the schema needs better depiction.

The supplementary material provided with the manuscript is easy to understand and corresponds to what is stated in the text. The only feature collected in the dataset but not mentioned in the manuscript is the presence of two different avatars (one male and one female). This feature should also be described in the text, along with information about the criteria used to select each participant's avatar. In addition, if possible, I suggest that the authors add more supplementary materials to make the experiment easier to analyse and reproduce. For instance, they can add images about the avatars used, or video regarding the behaviour of the avatar when touched.

Regarding the document structure, some sections and subsections may not be immediately clear. For instance, at line 195, it is unclear whether the section titled 'Experiment' is a standalone section or if it includes the sub-sections 'Hypothesis and Prediction', 'Conditions', 'Procedure', 'Participants', 'Measurement', and 'Environment'. I recommend that the authors better distinguish between the parts of the document, for example removing the “Experiment” section and integrating its subsections directly in the “Material & Method” part.
In addition, although useful as a summary of research’s purposes, the section “Hypothesis and Prediction” starting in line 197 is somewhat redundant as it repeats concepts already explained in the section “Haptic Feedback in Avatar Systems”.

Experimental design

The research described in the text is in line with the aim and scope of the journal considered. The analysis attempts to fill one of the knowledge gaps in the topic of interaction between interlocutor and tele-operated avatars, assessing the sensations felt by an operator when their tele-operated avatar is touched remotely. As reported in the manuscript, the study complies with ethical principles and was approved by the Ethics Committee of the ATR Review Board (501-4).

Although they can be inferred from the text, the research questions should be better explained and highlighted at the end of the Introduction and Conclusion section. In this regard, I suggest that the authors clarify about the types of perception they intend to investigate, both in relation to the interlocutor's touch (questionaries items Q1 – Q6) and in relation to the operator's perception of the avatar.

The experimental procedure is well described and generally easy to reproduce using the details reported in the text. If there is a weakness, it is that there is no information on the specifics of the devices used during the experiment (monitor, camera, etc...). In addition, there is no information on the modality used to share the videos between the different locations (remote, virtual, and local), as well as on the data transmission bandwidth capacity and the speed required to transfer data in real time.

Concerning the questionaries, I suggest the authors add more details about the items used to evaluate the operator’s perception of the avatar (intention to use and likeability). This information can be drawn by the supplementary material but does not figure out in the text, thus reducing the ability of the reader in understand the factors evaluated using the questionnaire.

Validity of the findings

The novelty contribution introduced by the paper is well discussed, as well as the gap it fills with reference to the current state of art. The data evaluated and discussed in the conclusions are provided with the article and can be easily used to reproduce the results reported in the “Results” section. The data were also anonymised in such a way that the identity of the participants in the experiment could not be determined.

Although the results reported are compelling, more details are needed with reference to the statistical analysis performed on the collected data. In particular, the authors should better describe how they have calculated the final values of “Intention” and “Likeability” used in two-way repeated measure ANOVA. Both the measures are evaluated using multi-items questionnaire (three questionary items for Intention, and five questionnaire items for Likeability), but there is no part in the text where is described how these items are summarized to generate a single value representative of the two aspects (intention to use and avatar’s likeability).

Finally, some considerations should be added in the conclusions about the cardinality of the data collected and employed in the study. Despite the relevant number of subjects involved, the data collected may not be enough to carry out meaningful statistical tests that allow us to draw general conclusions about the population considered. This factor should be taken into account in the analyses and should be discussed in more detail in the final chapter.

Additional comments

no additional comments

·

Basic reporting

- Besides some minor oversights, the manuscript presents a clear, unambiguous, and professional English language throughout. The found oversights are reported in what follows:
o Line 68: has inadequately been investigated.
o Line 350: refer consistently to the figures.
- The Introduction provides a sufficient level of details to understand the context in which the authors operate, however I would suggest adding a more structured presentation of the following sections and provide a clearer statement on the novelty of the authors’ proposal. This could also benefit from a clear description of the guiding research questions.
The background reports sufficient information to justify the authors’ investigation and choices.
- The structure almost conforms to PeerJ standards. The subheadings must be bold, followed by a period and the beginning of the new paragraph.
- Figures are well prepared and easy to understand.
- Raw data are supplied, and their interpretation is clear.

Experimental design

- I thank the authors for providing new insights on the presented topic. I think the provided investigation is at an early stage but is novel in the field of interest.
- While the main goal of the study is clear, i.e., understanding operators’ discomfort in relation to their avatar being touched, the research questions may be better highlighted to provide a better readability of the paper and interpretability of the results.
- The investigation is performed in a rigorous manner and clearly reports all the information required to ensure that it follows ethical standards. Having that the study received ethical approval, I would not argue that the participant’s name is asked in the questionnaire. However, this could represent a privacy-related issue.
o The methodology is sufficiently well described, allowing a possible replication of the results. I thank the authors for providing novel research on avatar operators discomfort by considering different “touching experiences”. However, when explaining the procedure, I suggest clarifying if the participants were informed of the possibility of having their avatar touched. If this is the case, the provided information could influence the operators’ reactions.

Validity of the findings

- While the study is at an early stage, it is novel and represents a good starting point for further analysis on the proposed topic. However, I would like to report some comments on the provided results and the possibility of considering gender and cultural related aspects.
o Considering the results, which I found particularly interesting, I would also have expected a finer analysis regarding the perceived differences between men and women, having that the perception of acceptable touch may vary among these groups.
o While the authors provide a first step towards the study of avatar operators’ discomfort, I would recommend the authors to provide some statements regarding the influence of gender and culture on the subjective perception of discomfort due to touch interactions. Even though they are not strictly related to the authors’ filed of application, I would suggest the following readings, which report interesting assessments on the perceived acceptability of affectionate touch depending on cultural aspects and gender, and the effects of human-robot touch. Notice that these are just suggested readings and I am not suggesting their inclusion in the manuscript.
Burleson, M. H., Roberts, N. A., Coon, D. W., & Soto, J. A. (2019). Perceived cultural acceptability and comfort with affectionate touch: Differences between Mexican Americans and European Americans. Journal of Social and Personal Relationships, 36(3), 1000-1022.
Zamani, N., Moolchandani, P., Fitter, N. T., & Culbertson, H. (2020, March). Effects of motion parameters on acceptability of human-robot patting touch. In 2020 IEEE Haptics Symposium (HAPTICS) (pp. 664-670). IEEE.
- The provided data are sufficient to reproduce the results and coherent with what has been reported in the manuscript.
- The conclusions could be extended to include all the previously made observations. In particular, providing more structured research questions in the introduction, may enhance the understanding of the concluding remarks.

Additional comments

I thank the authors for providing new insights on the proposed topic. I think it is particularly important to bring more analyses on operator perception and comfortability, to supplement the validity of avatar-based systems both from the operator and user perspectives.
While the proposed study seems to be at a very early stage, I think it represents a good starting point for further research on these topics. In particular, I would suggest considering more variables related to social interactions (human-human translated to human-avatar) when proposing other studies on the matter at hand.

Reviewer 3 ·

Basic reporting

no comment

Experimental design

no comment

Validity of the findings

no comment

Additional comments

The authors investigate an interesting issue: the effects on the operators of interactions involving touch between remote users and avatars. To this end, they design a proper task and evaluate the operator’s perceptions by questionnaires. Their findings indicate that operators can perceive a sensation of discomfort when their on-screen avatar is touched. Forty native Japanese speakers participated in the experiment. Do you think that cross-cultural differences may affect the main conclusions of the present study? It should be interested, as future work to compare the responses from participants from different countries/cultures. Please discuss this issue. Minor points: there are some acronyms not defined in the text, for example “CG”.

---

## Round 0.2 · accepted · Accept

The reviewers agree that the authors have addressed the previously reported problems. There is only one small typo that was reported by Reviewer 1, which the authors should correct. For this reason, I believe the paper is ready for publication.

Reviewer 1 ·

Basic reporting

In line 379, the reference to figure 7 is at the beginning of a sentence, so it should be reported as Figure 7 instead of Fig. 7.

Experimental design

no comment

Validity of the findings

no comment

Additional comments

no comment

·

Basic reporting

no comment

Experimental design

no comment

Validity of the findings

no comment

Additional comments

I thank the Authors for having addressed all the Reviewers' comments.
I have no further suggestions to make.

Reviewer 3 ·

Basic reporting

The authors have addressed satisfactorily to all reviewers' comments.

Experimental design

The authors have addressed satisfactorily to all reviewers' comments.

Validity of the findings

The authors have addressed satisfactorily to all reviewers' comments.